# Maize Autophagy-Related Protein ZmATG3 Confers Tolerance to Multiple Abiotic Stresses

**DOI:** 10.3390/plants13121637

**Published:** 2024-06-13

**Authors:** Mengli Liu, Li Ma, Yao Tang, Wangjing Yang, Yuying Yang, Jing Xi, Xuan Wang, Wanchao Zhu, Jiquan Xue, Xinghua Zhang, Shutu Xu

**Affiliations:** 1Key Laboratory of Biology and Genetic Breeding of Maize in Arid Area of Northwest Region, College of Agronomy, Northwest A&F University, Yangling 712100, China; 2646764970@nwafu.edu.cn (M.L.); mali0018@126.com (L.M.); 2022055150@nwafu.edu.cn (Y.T.); yang2294667587@nwafu.edu.cn (W.Y.); yyy17789365661@163.com (Y.Y.); 13934115695@163.com (J.X.); qfdwxzwc@163.com (W.Z.); xjq2934@163.com (J.X.); 2Yangling Qinfeng Seed-Industry Co., Ltd., Yangling 712100, China; wxwxkings@126.com

**Keywords:** autophagy, *ZmATG3*, abiotic stress, maize, Arabidopsis

## Abstract

Abiotic stresses pose a major increasing problem for the cultivation of maize. Autophagy plays a vital role in recycling and re-utilizing nutrients and adapting to stress. However, the role of autophagy in the response to abiotic stress in maize has not yet been investigated. Here, *ZmATG3*, which is essential for ATG8-PE conjugation, was isolated from the maize inbred line B73. The ATG3 sequence was conserved, including the C-terminal domains with HPC and FLKF motifs and the catalytic domain in different species. The promoter of the *ZmATG3* gene contained a number of elements involved in responses to environmental stresses or hormones. Heterologous expression of *ZmATG3* in yeast promoted the growth of strain under salt, mannitol, and low-nitrogen stress. The expression of *ZmATG3* could be altered by various types of abiotic stress (200 mM NaCl, 200 mM mannitol, low N) and exogenous hormones (500 µM ABA). GUS staining analysis of *ZmATG3*-GUS transgenic Arabidopsis revealed that *GUS* gene activity increased after abiotic treatment. *ZmATG3*-overexpressing Arabidopsis plants had higher osmotic and salinity stress tolerance than wild-type plants. Overexpression of *ZmATG3* up-regulated the expression of other *AtATGs* (*AtATG3*, *AtATG5,* and *AtATG8b*) under NaCl, mannitol and LN stress. These findings demonstrate that overexpression of *ZmATG3* can improve tolerance to multiple abiotic stresses.

## 1. Introduction

Most plants cannot move like animals when they are under stress; this causes their exposure to various abiotic stresses, which can have adverse effects on various plant developmental stages [1,2]. Plants have evolved some complex cellular pathways to enhance their adaptation to variable environments and their survival. Macro-autophagy (hereafter, ‘autophagy’) is an important pathway found in creatures in recent years [3].

In plants, autophagy is a conserved biological process [4] that plays a key role in defense against abiotic stresses [5]. The autophagy pathway of plants can be activated by a variety of abiotic stresses [6,7]. Overexpression of *MdATG18a* in apples (*Malus domestica*) increases autophagosome production, which enhances tolerance to drought stress [7]. Overexpression of *MsATG13* in alfalfa can enhance autophagy and antioxidant levels, which can increase plant cold tolerance [8]. In Arabidopsis, mutants of five autophagy-related genes, *ATG2*, *ATG5*, *ATG7*, *ATG9,* and *ATG10*, are sensitive to both salt and osmotic stress, and the performance of *GmATG8c*-overexpressing plants is greater than that of wild-type plants in germination assays [9]. *SiATG8a* improved the tolerance of Arabidopsis to both low-nitrogen and drought stress [10]. Autophagy is also very important in various other life activities, such as senescence [11,12,13], seed germination [14], and vegetative growth [3,15,16].

Autophagy begins with the formation of double-membrane-bound vesicles and autophagosomes, which sequester proteins, organelles that need to be degraded, and a portion of the cytoplasm to vacuoles or lysosomes for degradation [17,18]. A total of 36 evolutionarily conserved *ATG* genes were found in yeast cells, and most of these are required for the core autophagic machinery that generates the autophagosome [19]. These ATG proteins form four different functional complexes, which include the ATG1 kinase complex, ATG2/9/18 transmembrane complex, phosphatidylinositol 3-kinase (PI3K) complex, and two ubiquitin-like protein (ATG8 and ATG12) conjugation systems [20]. Two ubiquitin-like conjugation complexes, Atg12-Atg5-Atg16 and Atg8-phosphatidylethanolamine (PE), regulate phagophore expansion [19]. The ATG8-PE system contributes to the transport and maturation of autophagosomes, in addition to the specific targeting of autophagic contents [21]. The ATG8-PE conjugation system initially depends on a catalytic step mediated by ATG7, which is homologous to E1 ubiquitin-activating enzymes [22]. ATG7 activates ATG8 with a C-terminal glycine residue and transfers it to ATG3 (E2-like enzyme). Finally, ATG8 is conjugated to PE [23,24,25]. When cells lack the ATG12–ATG3 complex, mitochondrial mass is increased, and autophagosome targeting to the mitochondria is reduced. In addition, the absence of the ATG12-ATG3 complex increases the number of autophagosomes in cells under nutrient-rich conditions but not during starvation [26]. Previous studies have shown that overexpression of ATG3 improves the tolerance of plants to various abiotic stresses. For example, *MdATG3a* or *MdATG3b* improved the tolerance of Arabidopsis to mannitol or NaCl stress, and improved the growth of plants under nitrogen or carbon starvation stress [21]. In tea (*Camellia sinensis*) plants, *CsATG3a* has enhanced tolerance to nitrogen starvation [27]. However, it is not clear whether *ZmATG3* also plays an important role in abiotic stress.

To determine whether *ZmATG3* plays a role in response to abiotic stress, we characterized the functions of *ATG3* from maize, including its role in the response to abiotic stress, when it was overexpressed in yeast and Arabidopsis. Overexpression of *ZmATG3* confers tolerance to mannitol and salt stress and enhances growth performance under low-nitrogen (LN) conditions.

## 2. Materials and Methods

### 2.1. Plant Material and Treatments

The seeds of a maize inbred line (B73) were disinfected with 1% NaClO solution for 3 min, rinsed with distilled water four times, and germinated for 7 d. Then, they were transferred to a medium containing Hoagland nutrient solution and grown in a climate chamber (Ningbo Prantt Instrument Company, Ningbo, China, LRX-1100D-LED) at 28 °C, a relative humidity of 70%, and a light/dark cycle of 15 h/9 h.

To study the effects of treatment with NaCl, mannitol, and abscisic acid (ABA), three-leaf-stage seedlings were subjected to NaCl (200 mM) [28], ABA (500 μM), and mannitol (300 mM) [29]. The roots and leaves of seedlings were collected at 0, 3, 6, 9, and 12 h after stress. In order to simulate N starvation, we used Hoagland nutrient solution to culture maize seedlings. At the three-leaf stage, CaCl_2_ and KCl were used to replace Ca(NO_3_)_2_ and KNO_3_ in the nutrient solution [30]. The leaves were collected after 0, 12, 24, 36, 60, and 72 h after stress. The collected samples were frozen immediately in liquid nitrogen.

### 2.2. Sequence Analysis of ZmATG3

*ZmATG3* gene information was obtained from the Maize Genetics and Genomics Database (MaizeGDB, https://maizegdb.org/) and the National Center for Biotechnology Information (NCBI, http://www.ncbi.nlm.nih.gov/) database. The BlastP online tool was used to search for homologous protein sequences of different species in the NCBI protein database. The sequence of *ZmATG3* was aligned with homologous genes of other species using DNAMAN 7.0 software (Lynnon Biosoft, San Ramon, CA, USA). MEGA 11.0 software was used to generate the phylogenetic tree. NCBI and MEME (http://meme-suite.org/tools/meme, accessed on 14 June 2023) were utilized to predict the conserved motifs of *ZmATG3*. The STRING database (https://string-db.org/, accessed on 20 October 2023) was employed to make predictions of the protein interaction network. PlantCARE (http://bioinformatics.psb.ugent.be/webtools/plantcare/html/, accessed on 23 October 2022) was used to predict the cis-acting elements in the promoters.

### 2.3. GUS Activity Analysis

To determine whether the *ZmATG3* promoter responds to abiotic stress, we cloned the 2000 bp promoter of *ZmATG3*, ligated it into pUBI10-GUS (Beyotime Biotechnology, Shanghai, China), and transferred it into *Agrobacterium* strain GV3101 to generate ZmATG3-GUS transgenic Arabidopsis. Twelve-day-old *ZmATG3-GUS* transgenic Arabidopsis seedlings were transferred to 1/2 MS medium containing NaCl (100 mM) [31] and mannitol (200 mM) [32]. The first leaf of each plant was collected after 0, 12, and 24 h of treatment. GUS assays were carried out following standard protocols [33]. Arabidopsis leaves were stained using 5-bromo-3-indole-β-glucuronide at 30 °C. Finally, photographs were taken using a stereo fluorescence microscope (SMZ 25, Nikon, Tokyo, Japan). Primer information is supplemented in Appendix A (Schedule).

### 2.4. Expression of ZmATG3 in Yeast and Stress Tolerance Assays

To determine whether the ZmATG3 protein plays a role in yeast, the *ZmATG3* sequence was cloned into pYES2 (Weidi, Shanghai, China) to construct the pYES2-ZmATG3 vector. Then, the yeast strain BY4741 (MATa; his3∆1, leu2∆0, met15∆0, ura3∆0) was transformed with the recombinant yeast vector pYES2-*ZmATG3*. The coding sequences (CDS) of *ZmATG3* were recombined into the pYES2 vector (Invitrogen) via double digestion with KpnI and EcoRI. The pYES2-*ZmATG3* constructs and pYES2 empty vector were transformed into yeast BY4741 with the lithium acetate method [34].

After selecting recombinant colonies on SD-Ura plates, yeast cells with positive transformants were cultured in YPDA containing 2% galactose. To evaluate stress tolerance, they were diluted to different concentrations (10^0^, 10^−1^, 10^−2^, and 10^−3^) with an initial OD600 of 0.4, and a total of 5 μL of diluted bacterial droplets were added on the medium with 2% galactose containing or lacking NaCl (150 mM) and mannitol (500 mM) [35]; the mixture was then incubated at 30 °C for 3 days to monitor growth. For nitrogen deficiency stress, the cells were serially diluted (10^0^, 10^−1^, 10^−2^, and 10^−3^) with a uniform initial OD600 of 0.4, and a total of 5 μL of diluted bacterial droplets were added on SD-Ura agar medium with 2% galactose (without amino acids but plus 7.5 mM and 15 mM ammonium sulfate), which was cultured at 30 °C for 72 h.

### 2.5. Generation of Transgenic Plants and Phenotypic Analysis

To produce *ZmATG3*-OE lines, the *ZmATG3* sequence was cloned into pGreen-6 hemagglutinin (6HA) to construct the *ZmATG3*-6HA vector. Transgenic Arabidopsis plants were obtained by infecting Col-0 wild-type plants (WT) with *Agrobacterium tumefaciens*. Homozygous plants were screened on 1/2 × Murashige and Skoog (MS) medium supplemented containing 5 mg/L glufosinate ammonium, and two high-expression lines were used for further study. The complete list of primers utilized can be found in Appendix A.

The seeds underwent sterilization before being planted on 1/2 MS medium with varying concentrations of mannitol (0 mM, 200 mM) or NaCl (100 mM), and were cultivated in a greenhouse set at 23 °C. Following a 12-day germination period, the length of the primary roots was measured using a ruler. Rosette leaves were collected for real-time quantitative reverse transcription PCR (qRT-PCR) and enzyme viability analysis after 12 days of growth and stored at −80 °C. The experiments were replicated three times, with each repetition including a minimum of 10 seedlings.

### 2.6. Phenotypic Analysis of Transgenic Plants under Low-Nitrogen Stress

Seeds were sterilized before being planted in 1/2 MS medium inside a growth chamber (22 ± 1 °C and 16 h/8 h light/dark photoperiod). Fourteen-day-old Arabidopsis seedlings were transferred to hydroponic solution and grown in vermiculite. The plants received water for 3 days before being split into two groups for cultivation using LN (0.125 mM NH_4_NO_3_) and CK (2.5 mM NH_4_NO_3_) solutions. After 21 days, samples were taken from the roots and rosette leaves, and measurements were taken for rosette radius and leaf number. The dry matter content of plants was measured. A follow-up test of connecting-seat leaves was carried out.

### 2.7. Determination of Stress-Associated Physiological Indicators

Superoxide dismutase activity (SOD) (U/g) was determined by the NBT reduction method [36]; peroxidase activity (POD) (U/g) was determined by a guaiacol assay [37]; and catalase activity (CAT) (U/g) was determined by a hydrogen peroxide assay [37].

### 2.8. RNA Extraction and Quantitative RT-qPCR Analysis

Arabidopsis and maize (inbred line: B73) seedlings were subjected to RNA extraction utilizing the SteadyPure Plant RNA Extraction Kit (Accurate Biology, AG21019, Nashua, NH, USA). The cDNA was generated through the Evo M-MLV RT Mix Kit with gDNA Clean for qPCR Ver.2 (Accurate Biology, AG11728) as the template for real-time fluorescent quantitative PCR. Furthermore, we performed qPCR amplification on a QuantStudio 7 Flex instrument (ABI QuantStudio 7, Thermo Fisher Scientific, Waltham, MA, USA) according to the ChamQ SYBR qPCR Master Mix instructions (Vazyme, Q311-02, Nanjing, China). *AtActin8* and *Zmβ-Actin* were employed as internal reference genes for Arabidopsis and maize, respectively. Data from three biological replicates were analyzed using the 2^−∆∆CT^ method to determine mean values and standard deviations. The complete list of primers utilized can be found in Appendix A.

## 3. Results

### 3.1. Sequence Analysis of ZmATG3

We cloned *ZmATG3 from* maize inbred line B73, which was used as a reference because of its high-quality genome information in maize. There was only one *ATG3* gene in the B73 genome, and the full-length CDS of *ZmATG3* was 933 bp, which was deduced to encode a polypeptide of 311 amino acids. Sequence alignment of maize (ZmATG3) with Arabidopsis (AtATG3), rice (OsATG3), and wheat (TaATG3) revealed similarities of 69.94%, 88.92%, and 84.81%, respectively, in addition to the conserved HPC and FLKF motifs (Figure 1A). The motif of HPC, with the conserved cysteine residue (Cys253), is a putative active site necessary for autophagic E2 enzymes [25], and FLKF is a highly conserved motif. Then, the amino acid sequence of ZmATG3 was compared by homology alignment. The amino acid sequences of ten different species were downloaded from NCBI and phylogenetic analysis was carried out. The results suggested that the amino acid sequence of ZmATG3 was highly similar to that of ATG3 members in other plant species; ZmATG3 shared the highest similarity with SbATG3 in *Sorghum bicolor* (L.) (XP_021306485.1) (Figure 1B). Conserved domain prediction showed that these amino acid sequences of ATG3 in maize and other species encode *Autophagy_act_C* (Figure 1C). The conserved motifs and high homology of ZmATG3 and ATG3s in other species indicate that ATG3 proteins might be functionally conserved in different species. To obtain more information on *ZmATG3*, we used the PlantCARE online program to analyze the promoter. The 2000 bp sequences upstream of the “ATG” start codon of the *ZmATG3* gene were used to predict cis-acting elements. Many cis-acting elements associated with environmental stress were identified in the promoter region, including hormone-responsive elements (ABRE, AuxRR-core, CGTCA-motif, G-box, O_2_-site, TGACG-motif), stress-inducing elements (ARE), and light-regulatory elements (AAAC-motif, ATCT-motif, Box 4, GATA-motif, GT1-motif, I-box, TCT-motif), as well as some basic cis-acting elements, such as the TATA-box and CAAT-box (Figure 1D). This indicated that *ZmATG3* might play a role in the response to abiotic stresses.

### 3.2. Expression Analysis of ZmATG3 under Various Abiotic Stresses

To clarify the expression patterns of *ZmATG3* following exposure to stress conditions, B73 seedlings were subjected to various abiotic stresses at the three-leaf stage, including NaCl (200 mM), mannitol (300 mM), ABA (500 µM), and nitrogen starvation (0 mM) treatments. After treatment with NaCl, the expression of *ZmATG3* was up-regulated, and its expression was highest at 9 h (Figure 2A). *ZmATG3* expression levels also increased after mannitol stress treatment and exogenous ABA exposure (Figure 2B,C). Moreover, *ZmATG3* transcript levels increased 3.2-fold in the first 36 h of the treatment; smaller increases relative to the control were observed in the nitrogen starvation treatment (Figure 2D). These findings suggested that *ZmATG3* plays a role in the expression of responses to multiple types of abiotic stresses.

### 3.3. ZmATG3 Enhances the Tolerance of Yeast to Abiotic Stress

To determine the role of *ZmATG3* in the response to NaCl (150 mM), mannitol (500 mM), or LN stress (7.5 mM (NH_4_)_2_SO_4_; normal nitrogen: 15.0 mM (NH_4_)_2_SO_4_) in yeast, the pYES2-*ZmATG3* vector and the pYES2 empty vector were transformed into BY4741 yeast cells by PEG/LiAC. After *ZmATG3* was transferred into yeast, a spot test was performed to evaluate its function in yeast. The results showed that the transgenic yeast strains and the control strains could grow normally without stress (Figure 3). However, the growth of both types of yeast was inhibited under stress, including 500 mM mannitol and 150 mM NaCl. In contrast, transgenic yeast overexpressing *ZmATG3* had better growth and higher survival rates under NaCl, mannitol, or LN stress than under normal conditions.

### 3.4. Overexpression of ZmATG3 Enhances Tolerance to NaCl and Mannitol Stresses in Arabidopsis

To clarify the role of *ZmATG3* in abiotic stress, we used pUBI10 to transform the *ZmATG3* promotor linked to a GUS marker into the model plant Arabidopsis. The activities of the *ZmATG3* promoter under control (no treatment) and different stress conditions at 12 h and 24 h was detected by via GUS histochemical staining (Figure 4A). WT Arabidopsis leaves were not stained in the control or under NaCl (100 mM) and mannitol (200 mM) treatment. In contrast, the leaves of transgenic plants were darker blue after abiotic stress treatment compared with before treatment (Figure 4A). Next, the two overexpression lines were used to validate the function of *ZmATG3* (Figure 4B). To characterize the activity of the *ZmATG3* promoter under different stress conditions, the role of *ZmATG3* in the response to NaCl and mannitol stress was determined in Arabidopsis using the transgenic lines OE-7 and OE-10 (Figure 4B). When the seedlings of OE-7 and OE-10 were exposed to 200 mM NaCl for 14 days, both the *ZmATG3*-OE plants had better growth than WT plants (Figure 4C), as indicated by the fact that they had significantly longer roots than WT plants (Figure 4D). OE-7 and OE-10 also had significantly longer roots than WT plants (Figure 4C,D). SOD, POD, and CAT activities were measured for seedlings grown for 14 days. The results indicate that *ZmATG3*-OE plants had higher SOD, POD, and CAT activities than WT plants (Figure 4E–G). In general, these findings in *ZmATG3*-OE Arabidopsis plants suggest that *ZmATG3* could potentially enhance the tolerance of plants to different types of stresses.

### 3.5. Overexpression of ZmATG3 Enhances Tolerance to LN Stress in Arabidopsis

To determine whether *ZmATG3* is involved in the response to nitrogen stress, the growth and development of *ZmATG3*-OE plants were examined under both LN (0.125 mM NH_4_NO_3_) and control (2.5 mM NH_4_NO_3_) conditions. Under LN conditions, overexpressing plants had better growth than wild-type plants. For the maximum rosette radius, the overexpression lines showed an increasing trend compared with WT plants, and there were significant differences under LN conditions (Figure 5A,B). When grown under LN conditions, *ZmATG3*-OE plants had more rosette leaves than WT plants (Figure 5C). The roots of *ZmATG3*-overexpressing lines were longer than those of WT plants under LN conditions (Appendix A). The dry weight of roots and rosettes was significantly higher in *ZmATG3-OE* plants than in WT plants under LN conditions (Figure 5D,E). These results indicate that *ZmATG3*-OE plants have better adaptation in response to LN stress.

### 3.6. Overexpression of ZmATG3 Up-Regulated the Expression of Other AtATGs in Arabidopsis under Stress

Autophagy is a complex process, and nearly 40 *ATGs* are involved in this pathway. We determined the expression of the *ATGs* of Arabidopsis in *ZmATG3*-OE plants under multiple abiotic stresses, including the major autophagy genes *AtATG3*, *AtATG5,* and *AtATG8b* [38]. The expression levels of *AtATG3*, *AtATG5*, and *AtATG8b* in *ZmATG3*-OE plants were significantly higher than those in wild-type plants under 100 mM NaCl conditions (Figure 6A–C). Similarly, the expression of *ZmATG3* was significantly higher in *ZmATG3*-OE plants than in WT plants under 200 mM mannitol (Figure 6D–F) and LN conditions (Figure 6G–I). These results suggest that *ZmATG3*-OE plants can up-regulate the transcript levels of other important *AtATGs* under multiple types of abiotic stress. This might affect the growth and development of Arabidopsis.

## 4. Discussion

Autophagy is a recycling process that can enhance plant survival at the cellular and organismal levels in adverse environments. A number of autophagy-related (ATG) proteins have been cloned, especially ATG8 and ATG12, which play a key role in the autophagy pathway [39,40]. These ATG proteins are mainly involved in the formation of autophagosomes, double-membrane vesicles that capture and transport macromolecules to the vacuole for degradation [41]. So far, thirty-one core autophagy genes have been annotated from the maize genome, and one of them, ATG3, has been identified [42]. In this study, *ATG3* was cloned from maize B73. It was found that its nucleotide sequence was similar to that of *ATG3* in other plant species, and both had conserved HPC and FLKF motifs (Figure 1A). The HPC motif contains conserved cysteine residues. The conservation of this residue is characteristic of autophagic E2-like enzymes [23]. It indicates that *ZmATG3* is similar to the function of other eukaryotic cells and participates in the ATG8 conjugation system with a conserved mechanism.

In previous studies, many *ATG* gene expression levels were up-regulated and autophagy was enhanced under stress conditions [8,34,35,36,37]. After salt stress and drought stress, the expression of TaATG8s in wheat leaves was up-regulated and its autophagy activity was induced [38]. The expression of *SiATG8a* [10] and *MdATG9* [43] was strongly induced by drought and nitrogen starvation treatments. Likewise, we found that the expression of *ZmATG3* was induced by salinity (NaCl), drought (mannitol), ABA, and nitrogen starvation (Figure 2). GUS staining of transgenic Arabidopsis after mannitol and NaCl treatment further confirmed this result.

In addition, many autophagy genes can also enhance the tolerance of transgenic plants to various abiotic stresses. For example, *MdATG9* enhances the tolerance of transgenic apple calli to LN conditions [43], and *PagATG18a* significantly improves the salt tolerance of poplar trees [44]. Overexpression of *MdATG18a* in apples enhances alkali tolerance and heat tolerance, and overexpression of *MdATG10* in apples enhance salt and Cd tolerance [45,46,47,48]. Overexpression of *SiATG8a* in wheat confers tolerance to phosphorus starvation, and the overexpression of this gene in rice confers tolerance to nitrogen starvation [10,49]. Similarly, *ZmATG3* also enhances tolerance to salt, mannitol, and nitrogen starvation in transgenic Arabidopsis plants. *MdATG3a* and *MdATG3b* can complement atg3^−Δ^ yeast mutants under nitrogen starvation conditions [21]. In this study, *ZmATG3* was transformed into yeast, which improved the growth of yeast under mannitol, NaCl, and low-nitrogen conditions. Further study of the specific mechanism of *ZmATG3*-mediated tolerance to multiple stresses will help us deepen our understanding of the involvement of autophagy-related genes in abiotic stress in maize.

Autophagy can improve the antioxidant system by regulating the activity of antioxidant enzymes and the expression levels of antioxidant-related genes, thereby enhancing resistance to abiotic stress [50,51]. Enzyme defense systems that remove reactive oxygen species (ROS) include SOD, POD, and CAT, which help the body eliminate high ROS levels [52]. The SOD first converts oxide (O_2_^−^) into O_2_ and H_2_O_2_, which is then scavenged by CAT and POD [53,54]. Under salt stress, the SOD, POD, CAT, and APX activities of the *OxPagATG18a* line are higher than those of WT plants [44]. Apple plants overexpressing *MdATG8i* have higher activities of SOD and POD activities than WT plants under drought stress [55]. Under cold stress, the activities of CAT, POD, and SOD in *MsATG13* transgenic tobacco increased significantly [56]. H_2_O_2_ levels were significantly lower in ***M****sA**T**G1**3***-overexpressing plants than in WT plants after alkaline stress treatment [45]. Similarly, the activities of SOD, POD, and CAT in leaves of *ZmATG3*-overexpressing lines were higher than those of WT lines under NACL and mannitol stress (Figure 4E–G). This indicates that *ZmATG3* overexpression improves the antioxidant system (possibly through *ZmATG3*-mediated autophagy) and reduces ROS accumulation under mannitol and NaCl stress. Therefore, the yeast cell lines expressing *ZmATG3* showed better growth under stresses.

It has been reported that *ATG3* is an autophagy core gene, and overexpression of this gene may also promote the expression of other ATGs [27]. In fact, we found here that *ZmATG3* can induce the up-regulation of endogenous *ATGs* in transgenic plants under mannitol, salt, and low-nitrogen stress (Figure 6). ATG3 mainly catalyzes the transfer of ATG8 to PE and acts as a target for ATG12 coupling during autophagy [26,57]. This study found that the expression level of *AtATG3* in transgenic lines was significantly higher than that in WT under stress conditions (Figure 6A,D,G). As an important core protein, ATG5 is covalently coupled with ATG12 to participate in the autophagy pathway [50]. The expression level of *AtATG5* in transgenic lines was significantly higher than that in WT under stress conditions (Figure 6B,E,H). ATG8 is an important member of the ubiquitin-like coupling system and is responsible for the maturation of autophagosomes [58]. In addition, *AtATG8b* was up-regulated in both transgenic lines and WT, and the expression level was higher in transgenic lines under stress conditions (Figure 6C,F,I). This may be due to the overexpression of ZmATG3 leading to an increase in autophagy flux in Arabidopsis under stress, thereby increasing the expression of autophagy core genes. This is similar to the research results of others. For example, *MsATG13* promotes the production of autophagosomes in transgenic tobacco and enhances the cold tolerance of plants, which may have enhanced the antioxidant system [8].

In addition, autophagy can play a role in various physiological processes through interactions between its key genes and various cellular pathways [59]. The interacting proteins of ZmATG3 and AtATG3 were predicted using the STRING online website, and they were found to mainly interact with autophagy-related proteins (Appendix A). ATG3 can regulate resistance to abiotic stress by interacting with ATG-related proteins, and this interaction is enhanced in the presence of H_2_O_2_ in *N. benthamiana* [59]. Overall, *ZmATG3* can up-regulate the expression of other autophagy genes under abiotic stress, which can enhance the antioxidant enzyme system and thus abiotic stress tolerance. Therefore, it is a potential target for improving maize multi-stress resistance.

## 5. Conclusions

We found that *ZmATG3* is involved in the response to multiple abiotic stresses through overexpression in yeast and Arabidopsis. Heterologous expression of *ZmATG3* in yeast resulted in better growth under salt, drought, and LN stress. Furthermore, overexpression of *ZmATG3* in Arabidopsis increased tolerance to salt, mannitol-induced drought, and LN stresses based on the growth of whole plants. The results demonstrate that *ZmATG3* acts as a positive regulator in the response to multiple abiotic stresses such as salinity (NaCl), drought (mannitol), and LN. The regulatory effects of *ATG3* in maize require further study; such studies have implications for the use of *ZmATG3* to enhance responses to multiple abiotic stresses.

## Figures and Tables

**Figure 1 plants-13-01637-f001:**
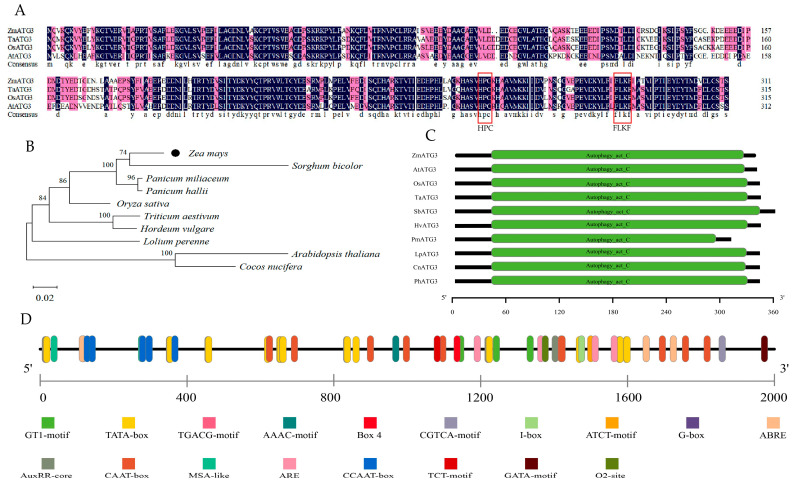
Sequence analysis of ZmATG3. (**A**) Amino acid sequence comparison of ZmATG3 with OsATG3 (KAB8080382.1), TaATG3 (ACN81639.1), and AtATG3 (NP_568934.1). Red boxes indicate HPC and FLKF motifs. (**B**) Molecular phylogenetic tree in the ZmATG3 protein and other ATG3 proteins. (**C**) Conservative domain analysis of ZmATG3 protein and other ATG3 proteins. (**D**) Cis-acting element analysis of *ZmATG3*.

**Figure 2 plants-13-01637-f002:**
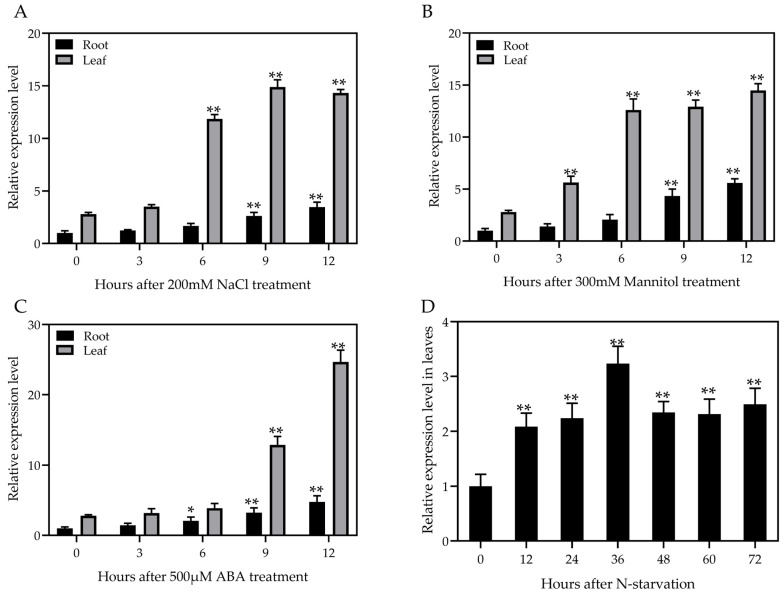
The expression pattern of *ZmATG3* under different abiotic stresses. Relative expression of *ZmATG3* in leaves and roots exposed to 200 mM NaCl (**A**), 300 mM mannitol (**B**), and 500 μM ABA (**C**), and leaves under N starvation treatment (**D**). Total RNA was extracted from leaves and roots in samples, and qRT-PCR was performed using gene-specific primers. Data are means ± SDs of three technical replicates. All significance tests were performed relative to 0 h, * *p* < 0.05, ** *p* < 0.01 (*t*-test).

**Figure 3 plants-13-01637-f003:**
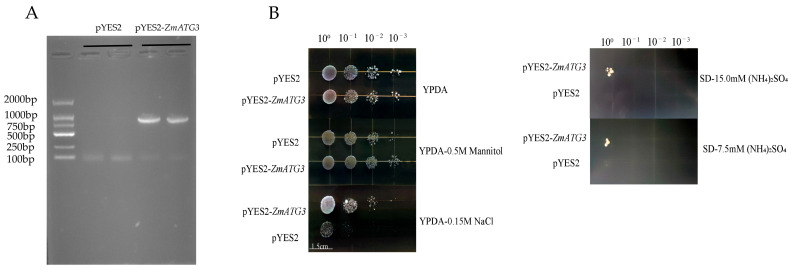
The growth activity of BY4741 (pYES2) and BY4741 (pYES2-*ZmATG3*) under different treatments. (**A**) Bacterial PCR; (**B**) growth of yeast under stress.

**Figure 4 plants-13-01637-f004:**
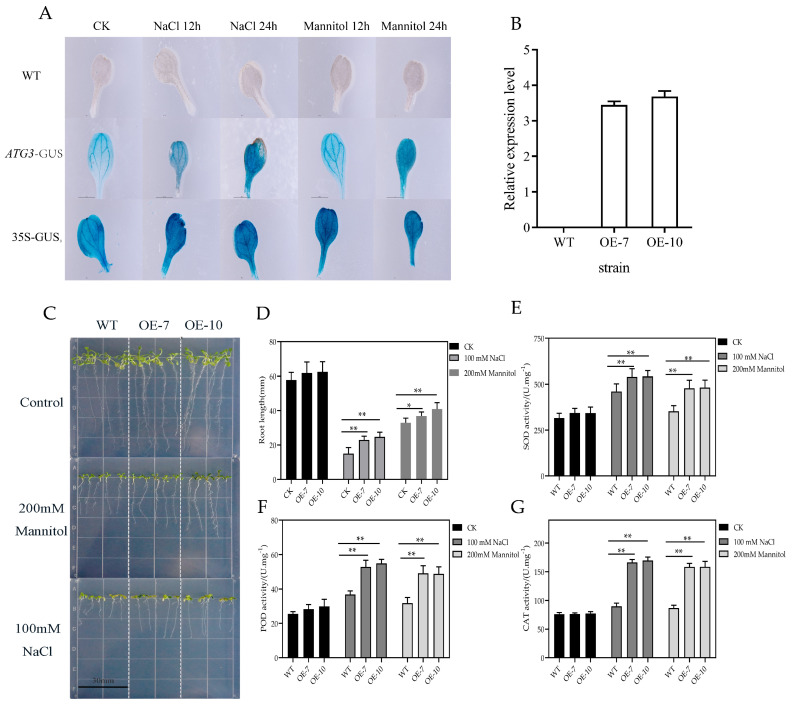
Phenotypes of transgenic Arabidopsis plants under NaCl and mannitol stress. (**A**) GUS staining in transgenic Arabidopsis plants under different abiotic stress treatments. (**B**) Screening and identification of overexpressing *ZmATG3* Arabidopsis plants. (**C**) Comparison of the growth of WT and OE lines under 1/2 MS control, 200 mM mannitol, or 100 mM NaCl treatment for 14 d. (**D**) Root lengths after treatment for 14 d. SOD activities (**E**), POD activities (**F**), and CAT activities (**G**) under 1/2 MS control, 200 mM mannitol, or 100 mM NaCl treatment for 12 h. Values are means ± SD (n ≥ 3). Asterisks indicate that there are significant differences. * *p* < 0.05, ** *p* < 0.01 (*t*-test).

**Figure 5 plants-13-01637-f005:**
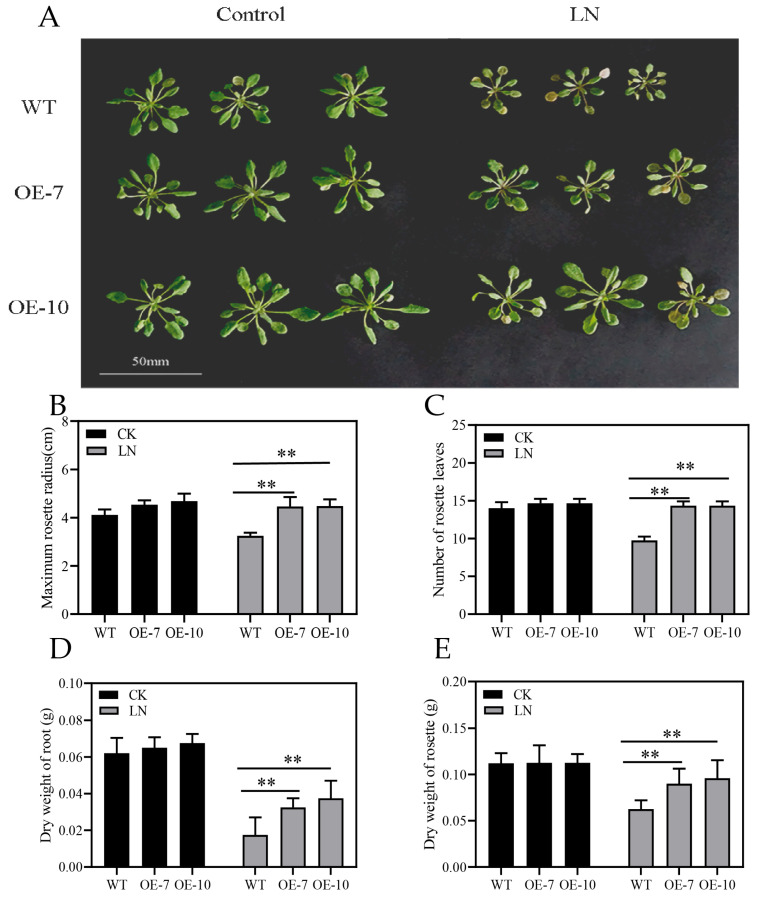
Comparison of the growth performance of *ZmATG3*-overexpressing (OE) and WT Arabidopsis plants. The rosette (**A**), maximum rosette radius (**B**), leaf number (**C**), dry weight of roots (**D**), and dry weight of the rosette (**E**) in OE and WT Arabidopsis plants under control and LN treatment for 21 d. Values are means ± SD (n ≥ 3). Asterisks indicate that there are significant differences. ** *p* < 0.01 (*t*-test).

**Figure 6 plants-13-01637-f006:**
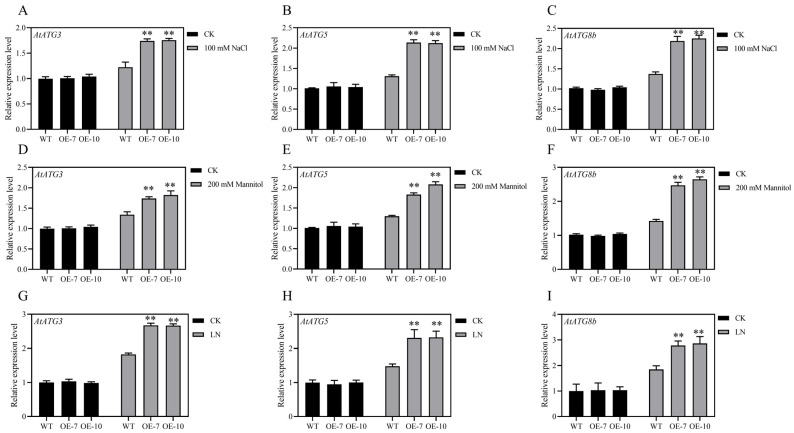
Transcript levels of genes involved in autophagy in Arabidopsis under different types of abiotic stress. Expression levels of (**A**) *ATG3*, (**B**) *ATG5*, and (**C**) *ATG8b* in rosette leaves under 1/2 MS control and 100 mM NaCl conditions. Expression levels of (**D**) *ATG3*, (**E**) *ATG5*, and (**F**) *ATG8b* in rosette leaves under 1/2 MS control and 200 mM mannitol conditions. Expression levels of (**G**) *ATG3*, (**H**) *ATG5*, and (**I**) *ATG8b* in rosette leaves under 1/2 MS control and LN conditions. Asterisks indicate that there are significant differences. ** *p* < 0.01 (*t*-test).

## Data Availability

The original contributions presented in the study are included in the article/Appendix A, further inquiries can be directed to the corresponding author.

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
