# Peer review of "Maize Autophagy-Related Protein ZmATG3 Confers Tolerance to Multiple Abiotic Stresses"

_plants, 2024, doi:10.3390/plants13121637_

Round 1

Reviewer 1 Report

Comments and Suggestions for Authors

The role of autophagy in maize has been less studied. In this manuscript, Liu et al. cloned ZmATG3 and explored its function under abiotic stresses by overexpressing ZmATG3 in yeast and Arabidopsis. Their findings indicate a positive role for ZmATG3 in responding to abiotic stresses. Overall, the manuscript is well-designed, and the majority of the data presented is persuasive. However, the following points should be addressed before publication.

1.    The results of N-starvation in yeast (Figure 3) are not entirely convincing. Only few clones grew under the 100 dilution condition, and none grew under further dilution conditions (10-1, 10-2 and 10-3). Additionally, the description “To determine how ZmATG3 protein contributes to response the …… or Nitrogen stress (Low nitrogen(LN):7.5mM (NH4)2SO4; Normal nitrogen: 15.0mM (NH4)2SO4) in yeast” in the manuscript is very confusing. Firstly, if 15.0mM (NH4)2SO4 is considered the normal growth condition for yeast, why do yeast cells hardly grow under this condition? Secondly, the difference in yeast growth shown in Figure 3 between 15.0mM (NH4)2SO4 and 7.5mM (NH4)2SO4 is minimal, making it challenging to conclusively determine that the overexpression of ZmATG3 improves yeast growth under low nitrogen exposure.

2. According to the expression patterns shown in Figure 2, ZmATG3 was highly expressed in leaves compared to roots and was also highly induced in leaves in response to abiotic stresses. Figure 4A shows ZmATG3 promoter-driven GUS expression in leaves. However, in Figure 4B, under NaCl and Mannitol stresses, the expression of ZmATG3 caused a significant improvement only in roots, with no significant difference observed in shoots compared to the wild type. This discrepancy needs clarification.

3.  The authors found that the expression of ZmATG3 can induce the upregulation of AtATG3/5/8b transcription levels. They should propose a mechanism in the Discussion section to explain how potential protein-protein interactions might promote this transcriptional upregulation.

4.    There are no supplementary figures in the files uploaded by the authors.

5.  In the abstract, it is mentioned that "the exogenous hormone (500 μM ABA)", but in the Methods and Results sections, the ABA concentration is stated as 100 μM. This inconsistency should be corrected for accuracy.

Comments on the Quality of English Language

The quality of English writing requires significant improvement. Here are a few examples from the Abstract section:

1.     “However, the role of autophagy in maize have not been investigated extensively, especially their response to abiotic stress” – “have” should be revised to “has.

2.     “In this study, ZmATG3, which is essential for ATG8-PE conjugation, were isolated from maize inbred line B73.” – “were” should be revised to “was.

3.     “GUS staining analysis in ZmATG3-GUS transgenic Arabidopsis showed significantly increasement of GUS gene activity after abiotic treatment.” – “significantly increasement” should be revised to “a significant increase in”.

4.     “Overexpress ZmATG3 can improve tolerance to multiple abiotic stresses.” – “overexpress” should be revised to “overexpressing”.

Authors should thoroughly revise the grammar mistakes in their manuscript.

Author Response

请参阅附件

Reviewer 2 Report

Comments and Suggestions for Authors

Introduction

Line 5: Please separate the sentence “years [3].In”

Line 6: Please remove the underscore.

Line 9: Add scientific name (Malus domestica).

Line 18: Replace "In summary, autophagy" with "Furthemore" or "also".

Line 18: Please separate the sentence “plants.Autophagy”

Line 41: Please add the scientific name to “tee plant”.

Line 43: Please complete this sentence (“However, …remains unclear”) and links to the main objective of your work.

Line 50: Add ABA.

Line 50-55: This information (“These results…abiotic stresses”) is not relevant in the introduction section. Please include in the abstract or conclusion.

Line 55: Please check this section.

2.1. Plant material and treatments

Line 3: Please include manufacturer name and model of chamber.

Line 11: Nitrogen? Please check the style of the Plants journal.

 2.2. Sequence analysis of ZmATG3

Line 3: Please check this "BLASTp". I think it must be "BlastP".

Line 5: Please include the reference (DNAMAN software).

 2.3. GUS activity analysis

Line 2: Please indicate the name of the manufacturer, city, state and country to pUBI10-GUS.

Line 3: Check if you should italicize. Please see the instructions in the Plants journal

Line 4: Italics?.Please check the style of the Plants journal.

Line 5: NaCl (100 mM) [32] or mannitol (200 mM). Please defines if it is “and”/”or”.

Line 7: Please separate the sentence “protocols [34],Arabidopsis”.

Line 9: Please include manufacturer name of fluorescence microscope.

Line 9: Suplemtnary Table 1 or Table S1?. Please determine this information.

2.4. Expression of ZmATG3 in yeast and stress tolerance assays

Line 2,3: Where is the recombinant yeast vector? Please indicate the name of the manufacturer, city, state and country.

Line 6: Please, add the citation to “lithium acetate method”.

Line 15: Please define if it is and/or.

2.5. Generation of transgenic plants and phenotypic analysis

Line 2: Please include the name of the vector manufacturer, city, state and country

Line 4: Italics are necessary. Please check the style of the Plants journal.

Line 5: Please include the name of the manufacturer or author

Line 8: What was the temperature of the greenhouse?

Line 8: Add days (12 days)

Line 10: Add the name of the qRT manufacturer, city, state and country

Line 13-20: There is no real connection with the previous paragraph. Is this a new section (2.6)? Demonstrate the connection based on the molecular and physiological evidence reported in your article.

2.7. RNA extraction and quantitative RT-qPCR analysis

Line 1: Here the correct thing should be "and" not "or"

Line 8-9: Please confirm this information. You used the reference gene ZmaACTIN or the β-actin mRNA gene?. This is not clear from the results.

3. Results

3.1. Sequence analysis of ZmATG3

Figure 1A: Please you should indicate this (boxes indicate HPC and FLKF motifs) in the alignment.

Line 34: Please complete this sentence at the end (These findings suggested that ZmATG3 plays a role in expression of response to multiple types of abiotic stresses)

3.2. Expression analysis of ZmATG3 under different types of abiotic stress

Figure 2D: The name of the sample (leaves or other) is required

Figure 2: “Expression levels were calculated relative to the expression of maize β-actin mRNA”. It`s not clear in the methodology, please check.

4. Discussion

Line 76: Please check the full name (N. benthamiana)

Line 81: Please check this section (“5. Conclusions”)

Comments on the Quality of English Language

English language fine. However, a minor revision is required. 

Reviewer 3 Report

Comments and Suggestions for Authors

This paper describes the relationship between the expression of Zea mays ATG3 and plant tolerance to stress conditions. The authors report that ATG3 expression is altered upon NaCl, low N, mannitol, and ABA treatments, and that these treatments modulate the activity of the ATG3 promoter. In addition, the authors found that transgenic plants constitutively expressing ATG3 show higher expression levels of some other ATG3 genes and a certain degree of tolerance to osmotic and salinity stress. The paper contains a large amount of experimental data and I think it would be of interest to the readers of Plants. Therefore, the paper is worth publishing. However, I have some concerns that should be addressed before the paper can be accepted for publication.

1. Page 7, 1st paragraph - "we transform ZmATG3 with the 35S promotor and GUS marker into the model plant Arabidopsis" - This sentence does not provide information on the construct used for plant transformation, and therefore the experiments described below this sentence are puzzling to the reader. If the construct contained the GUS gene under the control of the 35S promoter, why are the resulting plants called "ZmATG3 overexpressing"? If the GUS gene is under the control of the ZmATG3 promoter, what is under the control of the 35S promoter? The construct used for plant transformation should be described in detail.

2. For the ZmATG3-GUS transgenic lines described in the paper, it would be interesting to see any information on the tissue/organ-specific activity of the promoter. Were any changes in such specificity observed under stress conditions?

Comments on the Quality of English Language

Minor editing of English language required.

Author Response

请参阅附件
